# Psychological Well-Being of Malaysian University Students during COVID-19 Pandemic: Do Religiosity and Religious Coping Matter?

**DOI:** 10.3390/healthcare9111535

**Published:** 2021-11-10

**Authors:** Aisyah Che Rahimi, Raishan Shafini Bakar, Mohd Azhar Mohd Yasin

**Affiliations:** Department of Psychiatry, School of Medical Sciences, Universiti Sains Malaysia, Kubang Kerian 16150, Malaysia; aisyah.cr@usm.my (A.C.R.); raishanshafini@usm.my (R.S.B.)

**Keywords:** COVID-19, psychological disorder, university student, religiosity, religious coping

## Abstract

The COVID-19 pandemic and the restrictions imposed that changed the teaching and learning activities may add a psychological impact to the existing academic stress faced by university students. Past studies have associated low levels of psychological disorder with high religiosity and positive religious coping (RC). This study aimed to determine the level of psychological disorder among university students in Malaysia during the COVID-19 pandemic and measure their association with religiosity and religious coping (RC). An online cross-sectional survey was conducted between March and June 2020 involving 450 students. The survey instruments consisted of sociodemographic proforma, Duke University Religious Index (DUREL) for religiosity, Brief RCOPE Scale for RC and General Health Questionnaire-12 (GHQ-12) for psychological disorder; 36% of the participants experienced psychological disorder. Younger age, being a Muslim, living in the Green/Yellow zone and higher negative RC were significantly associated with psychological disorder. Higher positive RC was found to be protective against psychological disorder. However, the level of religiosity had no significant association with psychological disorder. In conclusion, the level of psychological disorder among university students has been high during the pandemic. Measures and interventions focusing on positive RC and reducing negative RC are recommended to improve the psychological well-being.

## 1. Introduction

The COVID-19 pandemic has a global impact, affecting not just physical health but also psychological well-being. The changes and restrictions imposed have taken a significant toll on people’s mental health [1,2]. As for university students, the changes and restrictions include changes from face-to-face to virtual teaching and assignments, and postponement or cancellation of classes and examinations. Online teaching and assessment may impose stress on students as many who have undergone home quarantine may not be fortunate enough to have access to the internet at home.

Previous research from throughout the world had found that university students had a high prevalence of psychological disorders and mental health problems [3,4,5]. A systematic review of 13 articles on Malaysian university students reported prevalence of stress and other psychological distresses up to 56% [6], higher than the general population’s prevalence of 29% [7]. During the COVID-19 pandemic, Cao et al. observed that 25% of undergraduate medical college students in China experienced anxiety symptoms [8]. Another study in France found the same tendency, with student participants reporting greater anxiety and moderate to severe stress [9]. Being a student was reported to be a significant positive predictor for depression compared to other employment status during the COVID-19 pandemic [10,11].

The impact of students’ psychological distress on academic performance and personal relationships has been systematically examined. Psychological distress had been associated with reduced mental capacity for academic activities, lower academic achievement and higher drop-out rates [5,12,13]. Students’ social and personal relationships were likely to be affected [12,13] with risk for suicidal thoughts and behaviours [14]. A survey conducted by the World Health Organization (WHO) on college students also revealed that mental disorders were predictive for impairment in multiple domains, including house chores, college/work-related tasks, interpersonal relationships, and the social domain [15].

For the past three decades, experts have been interested in the role of religion in health because it plays such a significant part in all aspects of human life [16]. In Malaysia, 99.3 percent of the population claimed to be religious [17], indicating that religion is an essential societal component. Scholars acknowledged that religion is a multifaceted construct that has been described in a variety of ways and forms in order to better understand how it contributes to health [18]. Early studies on the subject mostly examined religiosity using global indicators such as religious activity, religious affiliation, and subjective religiousness perception. Later, another dimension was investigated by assessing religious orientation, closeness to God, and attitudes toward the place of worship, as well as religious attitudes and beliefs [19]. In 1995, a National Institute on Aging and the Fetzer Institute conference on Methodological Approaches to the Study of Religion, Aging, and Health identified three major dimensions of religiosity or religious involvement; (1) Participation in public religious activities, such as attending religious services or participating in other group-related religious activities (prayer groups, Scripture study groups, etc.), is an example of organisational religious activity (ORA), (2) Non-organizational religious activity (NORA) involving religious activities performed in private, such as prayer, scripture study, watching religious shows or listening to religious channels, and (3) Intrinsic religiosity (IR) assessing degree of personal religious commitment or motivation [20].

Due to the multidimensional nature of religion, the relationship between religiosity and mental health was complicated and the findings were inconsistent. NORA and IR had a significant weak negative association with level of distress in a study of Malaysian mental patients, but ORA had no significant relationship with the latter [21]. Only IR was found to be negatively linked with depression in a study of medical students [22]. There were also some research works that revealed no link between religiosity and psychological well-being [23,24].

People will utilise various coping mechanisms to overcome the distress caused by a stressful event, in this case the COVID-19 pandemic, and the event outcomes are heavily influenced by the methods of appraisal and coping with the occurrences [19]. Religious coping is defined as abilities to comprehend and cope with life challenges through behaviours, emotions, relationships, and cognition that are linked to holy or supernatural forces. For the creation of Brief RCOPE questionnaires, Pargament established two categories of religious coping: positive and negative religious coping. Religious forgiveness, looking for spiritual support, collaborative religious coping, spiritual connection, religious purification, and compassionate religious reappraisal were the positive patterns signifying a secure connection with the scared, while spiritual dissatisfaction, punishing God reappraisals, and interpersonal religious discontent were the negative patterns reflecting the struggle and tension with the holy or sacred [19]. Each of these coping techniques had various effects on one’s health and mental well-being. Positive religious coping was linked to better psychological well-being [25,26] and positive psychological adjustment to stress [27], whereas negative religious coping was associated to religious strain and struggle [28] and had a detrimental impact on mental health [21,22,27,29].

As university students are the country’s greatest assets, this study aims to determine their psychological well-being during the COVID-19 pandemic, as well as the factors that contributed to it, such as religiosity and religious coping, as religion is an important component of Malaysia’s social structure.

## 2. Materials and Methods

This study was a cross-sectional study conducted from March until June 2020, which was during the full lockdown of the first wave of the pandemic in Malaysia. A set of online self-administered questionnaires were distributed among the undergraduate students of the three faculties (Medicine, Dentistry and Health Sciences) on the Health Campus, Universiti Sains Malaysia through students’ official university email and courses coordinator. The inclusion criteria were undergraduate students aged 18 years and above. Foreign students were excluded from this study. Based on the single mean formula and standard deviation of 2.56 from the previous study [30], the minimum sample size for the study was 400. This study was approved by the Human Research Ethics Committee of USM (JEPeM).

The online questionnaires distributed consist of pre-structured sociodemographic proforma designed for the study, General Health Questionnaire-12 (GHQ-12), Duke University Religious Index (DUREL) and Brief Religious Coping Scale (Brief RCOPE) used to assess the psychological well-being, level of religiosity and religious coping respectively. The questionnaires were provided in English and the Malay language, which the respondent could choose to answer in either language.

In the sociodemographic proforma, we included age, gender, ethnic groups, marital status, year of study, courses, educational funding, monthly household income and presence of any medical/psychiatric illness. For monthly household income, the respondents were categorized into three tiers according to Department of Statistics Malaysia; B40 represents the bottom 40% with income of less than RM4850/USD1166, M40 represents the middle 40% with income of RM4850-10959/USD1166-2635, whereas T20 represents the top 20% of Malaysian household income (>RM10960/USD2635) [31]. Some students were quarantined in the hostel, away from their families, while others stayed at home during the lockdown. The specific areas or districts in Malaysia were also classified into a few zones based on the COVID-19 active cases for the past 14 days, i.e., Red Zone if there were more than 40 active cases, Orange if there were 21 to 40 active cases, Yellow if there were one to 20 active cases, and Green if there were none at all [32].

### 2.1. Instruments

#### 2.1.1. General Health Questionnaire (GHQ-12)

The 12-item version of the General Health Questionnaire (GHQ-12) is used as a screening tool to detect the presence of psychiatric disorder [33]. Its validity is well established with high internal consistency (Cronbach’s alpha = 0.852) [34]. The study used binary scoring method whereby the two least symptomatic answers score 0 and the two most symptomatic answers score 1—i.e., 0-0-1-1. The minimum GHQ-12 total score was 0 and the maximum GHQ-12 total score was 12. The higher the total scores indicate poorer psychological well-being. Consistent with the original scoring system, the total score of 3 or below 4 should be classified as non-case, while scores of 4 and above indicates ‘caseness’ [34].

#### 2.1.2. Duke University Religious Index (DUREL)

The five-item DUREL questionnaire was used to measure the religious commitment of study participants [20]. The five items measure religious involvement in three different dimensions as mentioned before; (1) organization religious activity (ORA-1 item), (2) non-organizational religious activity (NORA–1 item), and (3) intrinsic religiosity (IR-3 items) [20]. Response options range from 1 (never) to 6 (more than once a day) for ORA, from 1 (never) to 6 (more than once a week) for NORA and from 1 (definitely not true) to 5 (definitely true of me) IR subscales. The higher the score in each subscale represented higher level of religious involvement or religiosity in respective dimension.

Its psychometric properties have been established to be reliable and valid to measure religiosity. The Malay version was validated by Nurasikin et al. in 2010 with moderate test-retest reliability (Spearman’s rho = 0.68) and fair internal consistency (Cronbach’s alpha: 0.45) [35].

#### 2.1.3. Brief Religious Coping Scale (Brief RCOPE)

The 14-item Brief RCOPE has two subscales, which measure the positive and negative religious coping strategies used by the subjects. The items consisted of seven positive coping items (P COPE) and seven negative coping items (N COPE). The inventory assesses degree to which the participants engage in both positive (established a sense of connectedness, and a secure relationship with a higher power or God) and negative religious coping (participants’ religious or spiritual struggle, tension, and conflict with a higher power or religious congregation) by using the four-point Likert Scale (1 = strongly disagree to 4 = strongly agree). The higher the scores for each subscale signified the higher degree of engagement in the type of religious coping strategy. Brief RCOPE has good internal consistency throughout study across different populations [19] and had been validated in the Malay language with Cronbach’s alpha of 0.87 for PCOPE and 0.88 for NCOPE [36].

### 2.2. Data Analysis

Data analysis was performed using the Statistical Package for Social Study Version 26 (SPSS 26.0). Descriptive statistics were used to describe the sociodemographics of the participants, level of religiosity, religious coping strategies and psychological disorder. Pearson correlation analysis was performed to explore the correlation between religiosity subscales, religious coping subscales and psychological disorder. Binomial logistic regression analysis was carried out to determine the significant factors associated with the psychological disorder of the participants. First, the independent variables, the sociodemographic data, ORA, NORA, IR, P COPE and N COPE, were screened using simple logistic regression. Some variables from sociodemographic data were further dummy coded to minimize uneven data distribution in univariate and multivariate analysis (Religion = Islam vs. non-Islam, Ethnic = Malay vs. Non-Malay, Funding = Self vs. Scholarship/Loan, Monthly Household Income = B40 vs. M40/T20). The variables with *p*-value < 0.25 were then selected for multiple logistic regression. Using Enter, Forward LR and Backward LR selection methods, variables with *p*-value of less than 0.05 were included in the final model. The forward LR method was chosen as the final model as it gave the best fit model.

## 3. Results

A total of 459 students out of 1955 completed the online questionnaires, resulting in a 23.4% response rate. Nine of them were foreign students, hence they were excluded from the study. All completed responses were included in the analysis.

### 3.1. Sociodemographic Characteristics

The sociodemographic characteristics of the 450 participants are summarized in Table 1. The participants were mainly female students (81.1%), with the average age of 21.85 years (SD: 1.892). The majority of the participants of this study made up of Malay students (73.1%), followed by Chinese (17.1%), Indians (7.1%), Bumiputras (2.7 %) and other ethnic groups (2.7%). Consistent with that, the majority of participants were reported as being Muslim (75.3%), while other religions like Buddhists, Christians, Hindus and others were reported by 12.2%, 6.4%, 4.9% and 1.1% of participants respectively.

Almost all (99.1%) of the students were single. Nearly half of them (48.4%) were health sciences students, whereas medical and dental students represented 26.2% and 25.3% of the participants. Majority of the students (54.9%) came from lower income background (B40) category. As for education funding, 82% of the students received financial aid from loan or scholarship while the rest were self-funded. 7.6% of the respondent had reported of having medical illness.

During this pandemic, 79.8% of the students were quarantined in the university hostel. According to the COVID-19 zones, 35.3% of students stayed in the Red and Orange Zones, while another 64.7% were in the Yellow and Green Zones at the time of the survey. More than half of the respondents (66.2%) indicated concern about academic changes during the pandemic, 44.9% of students were concerned about their family’s health, 33.6% were concerned about money, and only 26.3% were concerned about their own health.

### 3.2. Psychological Disorder, Level of Religiosity, and Religious Coping

Table 2 shows the mean, standard deviation and frequency of the GHQ-12 scores, level of religiosity and religious coping. The finding on the GHQ-12 reveals that 36.0% (*n* = 162) of students scored 4 or higher which indicates the presence of psychological disorder during the COVID-19 pandemic.

Concerning the religious involvement or religiosity, the students in this study show high religious involvement as the mean score is more than half of the maximum score for each subscale of DUREL. The mean score for ORA was 3.97 (SD = 1.271) out of 6, for NORA it was 4.68 (SD = 1.701) out of 6, and for IR it was 13.63 (SD = 2.19) out of 15 of maximum score.

The score of religious coping, as measured by the Brief RCOPE shows that most of the students practiced positive religious coping strategies more than the negative religious coping activities with mean PCOPE score of 23 out of 28 and mean NCOPE score of 10.43 out of 28.

### 3.3. Correlation between Religiosity Subscales, Religious Coping Subscales and Psychological Disorder

Pearson correlation analysis was applied to explore the correlation between religiosity subscales (ORA, NORA and IR) and religious coping subscales (PCOPE and NCOPE) with psychological disorder (Table 3). It was found that all the subscales had significant correlation with psychological disorder except for NORA. ORA (r = −0.12), IR (r = −0.099) and PCOPE (r = −0.128) were negatively correlated with psychological disorder while NCOPE (r = 0.209) was positively correlated with it though all the effects sizes were small.

### 3.4. Predictors for Psychological Disorder during the COVID-19 Pandemic

Table 4 showed simple and multiple logistic regression analysis respectively, exploring the association of sociodemographic profiles, level of religiosity, and religious coping with psychological disorder (GHQ-12 score of 4 and above). Age, religion, zone, positive religious coping and negative religious coping were found to have significant association with presence of psychological disorder after multivariable analysis. With an increase in the age of the participants by one year, there will be a decrease in the odds of psychological disorder by 12.8%. Non-Muslim students were observed to have 0.613 less odds to have psychological disorder than Muslim students. Those who lived in the Yellow or Green Zones during COVID-19 pandemic had 1.6 times higher odds of psychological disorder compared to those who lived in the Red or Orange Zones. In terms of religious coping, students with increased in NCOPE score by 1 had 1.17 times odds of psychological disorder in contrast with positive religious coping. An increase in the PCOPE score by 1 is associated with 0.216 lower odds of psychological disorder.

## 4. Discussion

The aim of our research was to look into the psychological well-being of undergraduate university students during the COVID-19 pandemic and the factors that contributed to it. Female students accounted for 81% of the respondents, which can be explained by the proportion of female students in the population of around 70%.

In the present cohort of university students, during the COVID-19 pandemic 36% of university students were reported to have a psychological disorder, which is consistent with the findings in the general population around the world [2]. The prevalence was found to be greater than in a local study conducted among medical students in the same location during a non-pandemic period [37], indicating that the COVID-19 pandemic had a negative impact on the students’ mental health. Aside from the health risks, the students had to deal with significant changes in their academic lives. In this study, more than two-thirds of the students indicated concern about their studies. Frequent changes in academic regulations in response to the pandemic’s progression created a lot of uncertainty, which contributed to psychological disorder. The transition from face-to-face to virtual or online classes, as well as changes in the evaluation method, all contributed to the student’s dissatisfaction [38]. As the participants were majoring in health-related courses which involved a lot of practical and clinical sessions, these sessions were likely being postponed or cancelled due to the social distancing. This scenario would compromise their competencies and interrupted their graduation process which posed concern among students [39].

Students in a younger age group were found to be significantly associated with psychological disorder similar to other studies [40,41]. Surprisingly, students who lived in a zone with fewer COVID-19 cases (0–20 cases) were associated with psychological disorder when compared to students who lived in a zone with more than 20 cases, contrary to the findings of other studies, which found that those who lived in high epidemic regions [42] or closer to COVID-19 cases [43] experienced more stress. This can be explained by the fact that the Green and Yellow Zones are typically rural and suburban areas with a lower population density than metropolitan areas, posing a lower chance of COVID-19 infection transmission [44]. However, the internet connection in rural places is likely to be poorer than in urban areas, causing undue stress to students who rely heavily on the internet for their online lectures.

Our study also found that religion played a role in psychological disorder especially during this pandemic. Muslim students were found to be more likely than Non-Muslim students to feel psychological disorder, similar to another local study that evaluated anxiety symptoms [22], probably because all religious meetings were suspended during lock-down. Because Muslims attend prayers at the mosque at least five times daily, significantly more frequently than other religious affiliations, their regular religious practices were disturbed, causing psychological disorder in practicing Muslim students. A study by Lavric and Flere in 2010 found that attending religious services was the most effective shield for lowering anxiety levels, when compared to personal prayers [45]. Furthermore, membership in religious congregations, according to social viewpoints, adds to a sense of belonging and even social identity, both of which support greater mental health [46,47].

Our findings strengthened the evidence that religious coping (RC) plays an important role in psychological well-being, with different religious coping styles contributing differently [18]. Positive religious coping was found to be a protective factor against psychological disorder [25,46,47], whereas negative religious coping was linked to a higher risk of psychological disorder [21,22,25].

Positive religious coping practices created a sense of hope and optimism in dealing with the uncertainty and possible negative impacts brought on by the COVID-19 pandemic by having a strong link with a divine force and a conviction that there is a silver lining underneath all of this [18]. People who have a high level of positive religious coping also have a higher level of self-reflection and hopeful thinking, which leads to a lower sense of helplessness and reducing psychological disorder. Negative religious coping, on the other hand, poses a psychological risk by causing difficulties in the spiritual or religious path and producing a pessimistic outlook on life [18]. These coping mechanisms can be extremely distressing since they center on viewing crises, like the COVID-19 pandemic, as retribution from God, being abandoned by God, believing that the events’ effects are devilish acts, and questioning one’s relationship with the sacred. As a result, it is no surprise that poor religious coping techniques are linked to higher levels of psychological suffering.

Although the participants are considered moderate to highly religious based on the objective scale, this study failed to establish the association between religiosity and psychological disorder, which is similar to another local study on medical students [22]. This can be explained as the respondents come from multicultural and multireligious groups that may have different concept and understanding of religiosity. They may value spirituality, which is individualized human experience related to the sacred, rather than a formal religious belief, which is influenced by society or social group. Furthermore, because religion is a complex and multidimensional phenomenon, religiosity as measured by an objective scale may not reflect one’s genuine nature and engagement in religion.

This study highlighted the mental health of multireligious university students in Malaysia during the COVID-19 pandemic and the factors that contributed to it, with a focus on religiosity and religious coping. It gave insight on the relationship of religion and mental health that was not explored much in our local setting.

The main limitation of the study is its cross-sectional nature and its study population which was confined to university students taking health-related courses only. Thus, this may limit the generalizability of the result to other populations. Exploring the students’ concerns in depth, particularly on academic-related events, can provide more insight into how to modify the educational system to meet the unmet requirements of students during a pandemic. Because this study was conducted during the early stages of the pandemic, it would be fascinating to compare the findings with those obtained during the later stages of the pandemic to have a better understanding of the impact of the pandemic at different points of time.

## 5. Conclusions

In conclusion, the level of psychological disorder among university students during the COVID-19 pandemic is high with religious coping found to play a role in it. The findings of this study sanction the importance of taking measures to ensure the continuity of religious and spiritual activities during the pandemic. In addition, training of mental health care professionals should focus on enhancing the positive religious coping and improving negative religious coping among those affected by the COVID-19 pandemic.

## Figures and Tables

**Table 1 healthcare-09-01535-t001:** Sociodemographic background of the participants (*n* = 450).

Variables	Frequency, *n* (%)
Age	21.85 (1.892) *
Gender	
Male	85 (18.9)
Female	365 (81.1)
Ethnic	
Malay	329 (73.1)
Chinese	77 (17.1)
Indian	32 (7.1)
Bumiputra	12 (2.7)
Religion	
Islam	339 (75.3)
Buddha	55 (12.2)
Hindu	22 (4.9)
Christian	29 (6.4)
Others	5 (1.1)
Marital Status	
Single	446 (99.1)
Married	4 (0.9)
Year of Study	
1	146 (32.4)
2	100 (22.2)
3	70 (15.6)
4	85 (18.9)
5	49 (10.9)
Final Year	
Yes	84 (18.7)
No	366 (81.3)
Courses	
Medicine	114 (25.3)
Dentistry	118 (26.2)
Health Sciences	218 (48.4)
Funding	
Self-funding	81 (18.0)
Loan	220 (48.9)
Scholarship	149 (33.1)
Any medical disorder	
Yes	34 (7.6)
No	416 (92.4)
Monthly Household Income (RM/USD)	
B40 (<RM4850 /USD1166)	247 (54.9)
M40 (RM4850-10959/USD1166-2635)	166 (36.9)
T20 (>RM10960/USD2635)	37 (8.2)
Living Arrangements	
Hostel	359 (79.8)
Home	91 (20.2)
Zone (COVID-19 Local Spread)	
Red/Orange	159 (35.3)
Yellow/Green	291 (64.7)
Health	
Worry	118 (26.3)
Not worry	332 (73.7)
Academic	
Worry	298 (66.2)
Not worry	152 (33.8)
Financial	
Worry	151 (33.6)
Not worry	299 (66.5)
Family’s Health	
Worry	202 (44.9)
Not worry	248 (55.1)

* Mean (SD).

**Table 2 healthcare-09-01535-t002:** Level of psychological disorder, religiosity and religious coping of participants (*n* = 450).

Variables	Mean (SD)
General Health Questionnaire-12 (GHQ-12)	3.12 (3.316)
Caseness (Score ≥ 4)	162 (36) *
Non-case (Score < 4)	288 (64) *
Duke University Religious Index (DUREL)	
Organization Religious Activity (ORA)	3.97 (1.271)
Non-organizational Religious Activity (NORA)	4.68 (1.701)
Intrinsic Religiosity (IR)	13.63 (2.19)
Total	22.27 (4.157)
Brief Religious Coping Scale (Brief RCOPE)	
Positive Religious Coping (PCOPE)	23 (4.909)
Negative Religious Coping (NCOPE)	10.43 (3.145)

* Frequency, *n* (%).

**Table 3 healthcare-09-01535-t003:** Pearson correlation analysis between religiosity subscales, religious coping subscales and psychological disorder.

	ORA (DUREL)	NORA (DUREL)	IR (DUREL)	PCOPE	NCOPE	Psychological Disorder
ORA	1	0.311 **	0.310 **	0.316 **	0.053	−0.120 *
NORA	0.311 **	1	0.659 **	0.630 **	0.027	0.023
IR	0.310 **	0.659 **	1	0.767 **	−0.023	−0.099 *
PCOPE	0.316 **	0.630 **	0.767 **	1	0.083	−0.128 **
NCOPE	0.053	0.027	−0.023	0.083	1	0.209 **

* *p* < 0.05, ** *p* < 0.01.

**Table 4 healthcare-09-01535-t004:** Logistic regression analysis to determine predictors for psychological disorder.

Variables	Simple Logistic Regression			Multiple Logistic Regression		
	Regression Coefficient (b)	Crude OR (95% CI)	*p*-Value	Adjusted b	Adjusted OR (95% CI)	*p*-Value
Age (years)	−0.111	0.895 (0.80, 0.99)	0.049 *	−0.137	0.872 (0.775, 0.981)	0.023 **
Gender						
Male	0	1				
Female	0.298	1.35 (0.81, 2.23)	0.249 *			
Ethnic						
Malay	0	1				
Non-Malay	−0.227	0.797 (0.51, 1.24)	0.313			
Religion						
Islam	0	1		0	1	
Non-Islam	−0.318	0.728 (0.46, 1.15)	0.175 *	−0.95	0.387 (0.209, 0.716)	0.002 **
Marital Status						
Single	0	1				
Married	−20.64	0.000 (0.000)	0.999			
Final Year						
Yes	0	1				
No	0.557	1.746 (1.02, 2.96)	0.039 *			
Courses						
Medicine	0	1				
Dentistry	−0.385	0.68 (0.43, 1.07)	0.096			
Health Sciences	0.564	1.75 (1.19, 2.59)	0.004 *			
Funding						
Self-funding	0	1				
Loan/Scholarship	0.309	1.36 (0.83, 2.22)	0.217 *			
Any medical disorder						
Yes	0	1				
No	−1.014	0.36 (0.178, 0.74)	0.005 *			
Monthly Household Income						
B40	0	1				
M40/T20	0.386	1.471 (0.99, 2.16)	0.051 *			
Living Arrangements						
Hostel	0	1				
Home	0.249	1.28 (0.8, 2.05)	0.301			
Zone						
Red/Orange	0	1		0	1	
Yellow/Green	0.48	1.628 (1.07, 2.46)	0.021 *	0.474	1.606 (1.034, 2.493)	0.035 **
DUREL						
ORA	−0.2	0.819 (0.7, 0.95)	0.011 *			
NORA	0.02	1.029 (0.91, 1.15)	0.62			
IR	−0.09	0.91 (0.83, 0.99)	0.038 *			
Brief RCOPE						
PCOPE	−0.053	0.948 (0.91, 0.98)	0.007 *	−0.123	0.884 (0.839, 0.932)	<0.001 **
NCOPE	0.14	1.15 (1.07, 1.22)	<0.001 *	0.159	1.173 (1.096, 1.255)	<0.001 **

* Significant variables with *p* < 0.25 were included in the multiple logistic regression analysis. ** Variables with *p* < 0.05 were retained for the final model. For MLR, the Forward LR method was chosen as the final model. Classification table = 69.1%, Hosmer–Lemeshow test *p*-value = 0.677, area under ROC curve = 69.3%.

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
