# Peer review of "Psychological Well-Being of Malaysian University Students during COVID-19 Pandemic: Do Religiosity and Religious Coping Matter?"

_healthcare, 2021, doi:10.3390/healthcare9111535_

Round 1

Reviewer 1 Report

Dear Authors,

your paper entitled “Psychological Distress among Malaysian University Students during COVID-19 Pandemic: Do Religiosity and Religious Coping Matter?” is interesting, but I believe that it needs some revisions before it can be considered for possible publication, as for my comments listed here below for the different sections of the manuscript.

Best regards,

the Reviewer.

Abstract

  1. Lines 22-24 repeat the same concepts stated in the lines above (20-21)

Introduction

  1. General comment: the Introduction is perhaps too long. Please consider to shorten it, cutting at least one or two sentences.
  2. Line 28: not sure whether it is necessary to clarify “which was caused by the virus SARS-CoV-2”, I believe it is well known
  3. Lines 27-33: Unfortunately I believe you should use present tense and past continuous (e.g. the pandemic has an impact) as the phenomena are very recent and still very actual in terms of impacts on people.
  4. Lines 38-39: too much generic sentence. What kind of metal health problem(s)? Are you sure that it is proved that students are at an increased risk compared to other populations? Which populations? Consider taht many of the studies investigating mental health among students are cross-sectional surveys.
  5. Line 39-41: please limit your observations to the single studies and don’t generalize their findings on a national level.
  6. Line 45: don’t use higher risk, but perhaps “a factor positive associated with increased odds etc”
  7. Lines 57-58: again, please don’t make general assumption from a single study report, and be more precise on the data you are referring to.
  8. Line 60: is prayer a phrase?
  9. Line 70: when you refer to “church” you need to specify that the study was set in a Christian community.
  10. Line 77: replace Bible with Scripture
  11. Materials and Methods
  12. Please try to follow the Strengthening the Reporting of Observational Studies in Epidemiology (STROBE) guidelines. Many information are lacking in your M&M section https://www.equator-network.org/wp-content/uploads/2015/10/STROBE_checklist_v4_combined.pdf
  13. Line 130: Please, note that distress is a very specific concept. Please check whether the primary outciome of GHQ-12 is to measure psychologiucal distress (i.e. according to the original description of the questionnaire provided by the Authors), as you state, or perhaps psychological well-being or a general level for common psychological/psychiatric disorders.
  14. Lines 137-140: same observation as above.
  15. Section 2.1.2: perhaps slightly repeated concepts when compared top Introduction?
  16. Line 147 “motivation16”?
  17. Line 157: not sure what the “M” after BRCOPE stands for. Maybe it should be explained.
  18. Lines 173.174: what is the model used in the Multiple logistic regression analysis?
  19. Line 175: what covariates?

3.Results

  1. Line 181: the sentence is repeated compared to M&M section.
  2. Line 181: according to the results presented in the following lines, there are kind of a lot of “non-Malaysian” in the study group. Why these nine students have been excluded? Please clarify in the text the criteria.
  3. Line 184: perhaps “are” instead of “were”?
  4. Table 1: the central column is not needed, it is almost empty. Consider other solutions.
  5. Some words in Table 1 have an asterisk but in the footnote I find a number “1”.
  6. I don’t get the meaning of the footnote of the Table: perhaps the text needs to be moved to the M&M section?
  7. The variables related to the “Monthly Household Income (RM)” need to be specified and referred to an amount of money and compared to US dollars in order to be understood by an international audience.
  8. Line 212 and 217: why “was” peresented?
  9. Table 2: consider to revise according to the observation for M&M section.
  10. Table 3: Why “(score”? and only after “ORA”?
  11. Table 2 and 3: can be included in one Table only. Please make an effort to design a better Table.
  12. Pearson analysis not anticipated in M&M section.
  13. Line 241-2 “A student with 1 year 241 younger has 22.8% less likely to have psychological distress” not sure of having understood this sentence.

Discussion

  1. Please add a limitation section

Conclusions

  1. Lines 33-336: this is not a conclusion of your study.

References

  1. 55 references are a little bit too much for this kind of study: please consider to reduce the list to 50 at least, avoiding unnecessary citations.

Reviewer 2 Report

This is a well-conducted and analysed study. Apart from having a check on English expression, I only have a few suggestions. With respect to the English language, one example is the use of the word 'inspect' in the Abstract. Perhaps change this to 'assess' or 'measure'. I would also suggest that the use of 'skyrocketed' (line 59) and 'fascinated' (line 61, 323)(perhaps replace with 'interested/interesting') are not appropriate for a scientific report. You might also clarify 'backbone' (line 103) and 'journey' (line 309). See also line 137 re: "In consistent ...": should you delete the 'In'? In some cases where references are made to 'links' between variables (e.g., line 84, 'strongly linked'), it would be helpful to state whether the 'link' was a positive or negative correlation - and clearly articulate the relationship between variables (as in other parts of the paper).  I would suggest stating the number of eligible students and the response rate (459/1955) in Section 2.  You also need to comment on the fact that 81% were female: does that represent a significant bias in response - or does it reflect the gender ratio in the 1955 students? If a bias, then reporting on the gender variable needs to be heavily qualified, along with all the results. You might also comment on the lack of association between religiosity and distress as possibly reflecting the fact that religiosity was high in this sample. Overall, it would be helpful if you summarised the various limitations of the study in the Discussion.

Round 2

Reviewer 1 Report

Congratulations to the Authors as they have properly addressed all of my comments.